# Theoretical Study of the Thermal Rate Coefficients of the H_3_^+^ + C_2_H_4_ Reaction: Dynamics Study on a Full-Dimensional Potential Energy Surface

**DOI:** 10.3390/molecules29122789

**Published:** 2024-06-12

**Authors:** Tatsuhiro Murakami, Soma Takahashi, Yuya Kikuma, Toshiyuki Takayanagi

**Affiliations:** 1Department of Chemistry, Saitama University, Shimo-Okubo 255, Sakura-ku, Saitama City 338-8570, Japan; s.takahashi.144@ms.saitama-u.ac.jp (S.T.); y.kikuma.323@ms.saitama-u.ac.jp (Y.K.); 2Department of Materials & Life Sciences, Faculty of Science & Technology, Sophia University, 7-1 Kioicho, Chiyoda-ku, Tokyo 102-8554, Japan

**Keywords:** astrochemistry, interstellar medium, branching mechanism, full-dimensional potential, ring polymer molecular dynamics

## Abstract

Ion–molecular reactions play a significant role in molecular evolution within the interstellar medium. In this study, the entrance channel reaction, H_3_^+^ + C_2_H_4_ → H_2_ + C_2_H_5_^+^, was investigated using classical molecular dynamic (classical MD) and ring polymer molecular dynamic (RPMD) simulation techniques. We developed an analytical potential energy surface function with a permutationally invariant polynomial basis, specifically employing the monomial symmetrized approach. Our dynamic simulations reproduced the rate coefficient of 300 K for H_3_^+^ + C_2_H_4_ → H_2_ + C_2_H_5_^+^, aligning reasonably well with the values in the kinetic database commonly utilized in astrochemistry. The thermal rate coefficients obtained using both the classical MD and RPMD techniques exhibited an increase from 100 K to 300 K as the temperature rose. Additionally, we analyzed the excess energy distribution of the C_2_H_5_^+^ fragment with respect to temperature to investigate the indirect reaction pathway of C_2_H_5_^+^ → H_2_ + C_2_H_3_^+^. This result suggests that the indirect reaction pathway of C_2_H_5_^+^ → H_2_ + C_2_H_3_^+^ holds minor significance, although the distribution highly depends on the collisional temperature.

## 1. Introduction

The H_3_^+^ molecule plays a crucial role in interstellar chemistry due to its efficiency as a proton donor in H_3_^+^ + X → H_2_ + HX^+^ ion–molecule reactions, where X represents a neutral molecule in the interstellar medium [1,2,3]. Many interstellar molecules detected using spectroscopic techniques fulfill this proton affinity condition. Furthermore, these ion–molecule reactions occur efficiently due to the absence of an entrance reaction barrier, primarily driven by strong attractive forces like charge–dipole or charge-induced–dipole interactions. Consequently, H_3_^+^ had often been referred to as the initiator of interstellar chemistry in previous studies [1,3,4,5,6]. Ethylene (C_2_H_4_) also plays a role in interstellar chemistry since the molecule has been detected in the circumstellar envelope of the carbon-rich asymptotic giant branch star IRC+10216 [7,8,9,10]. Therefore, laboratory studies have been carried out to determine the rate coefficient for H_3_^+^ + C_2_H_4_ reactions [11,12,13,14]. H_3_^+^ + C_2_H_4_ reactions predominantly yield the H_2_ + C_2_H_5_^+^ product via a proton-donating process with considerable exothermicity [13,14]. Although this barrierless reaction appears to be straightforward, experimental investigations have yet to address the temperature dependence of the reaction rate coefficients [11,12,13,14]. While Langevin’s theory is commonly utilized for estimating the low-temperature rate coefficients in various ion–molecule reactions, calculating the reaction rate coefficients using first principles may be essential for assessing the validity of Langevin’s theory. From an astrochemical perspective, understanding the H_3_^+^ + C_2_H_4_ → H_2_ + C_2_H_5_^+^ reaction rate coefficients and their temperature dependence is quite important.

As mentioned above, this ion–molecule reaction yields H_2_ + C_2_H_5_^+^ predominantly. It is worth noting that the reaction H_3_^+^ + C_2_H_4_ → 2H_2_ + C_2_H_3_^+^ can lead to the formation of three species due to its exothermic nature. This reaction follows an indirect mechanism, where the C_2_H_3_^+^ product is formed from vibrationally excited C_2_H_5_^+^ via a unimolecular reaction, with vibrational energy provided in the initial proton transfer process from H_3_^+^. Thus, the process can be formally expressed as H_3_^+^ + C_2_H_4_ → H_2_ + C_2_H_5_^+^* → H_2_ + H_2_ + C_2_H_3_^+^, where C_2_H_5_^+^* represents a molecule with sufficient vibrational energy to produce the H_2_ + C_2_H_3_^+^ fragments. Experimental measurements using ion cyclotron resonance under room temperature and low-pressure conditions have yielded branching fraction values of 0.40:0.60 [13] and 0.31:0.69 [14] for [C_2_H_5_^+^]:[C_2_H_3_^+^], indicating that the C_2_H_3_^+^ ion is the predominant reaction product. It is worth mentioning that previous experimental studies have reported the lifetime of the primary C_2_H_5_^+^ reaction product to be in the range from 10^−8^ to 10^−6^ s [14]. This extended lifetime suggests that the indirect mechanism prevails in the C_2_H_3_^+^ production channel, which is consistent with previous quantum chemistry calculations conducted by Uggerud, who reported barrier and exit energy values of 55.9 and 50.0 kcal/mol (234 and 209 kJ/mol), respectively, at the MP2/6-31G(d,p) level of theory [15]. The overall rate coefficient at 300 K published in the KIDA (Kinetic Database for Astrochemistry) database is 2.9 × 10^−9^ cm^3^ s^−1^ [16], with a branching fraction of 0.30:0.70 for [C_2_H_5_^+^]:[C_2_H_3_^+^], as reported in the experiments mentioned earlier and conducted using a drift chamber mass spectrometer [17]. Based on the branching fraction results, the C_2_H_5_^+^* fragment acquires significant rovibrational energies from the H_3_^+^ + C_2_H_4_ proton-donating process, leading to its dissociation into H_2_ + C_2_H_3_^+^ fragments. Understanding the internal energy of C_2_H_5_^+^ in the H_3_^+^ + C_2_H_4_ → H_2_ + C_2_H_5_^+^ reaction dynamics and its temperature dependence is essential for elucidating the branching mechanism of C_2_H_5_^+^/H_2_ + C_2_H_3_^+^. Comprehending the branching mechanism as well as the thermal rate coefficient is critical in astrochemistry.

Motivated by this goal, we investigated the H_3_^+^ + C_2_H_4_ → H_2_ + C_2_H_5_^+^ reaction dynamics. Previously, the potential energy profiles of the entire C_2_H_7_^+^ system were studied using quantum chemical calculations [18,19,20]. These studies identified several equilibrium structures of the C_2_H_7_^+^ potential energies, with the proton-bridged (CH_3_-H-CH_3_)^+^ structure and its associated forms being the most stable [18,19,20]. While the dissociation channels of the C_2_H_7_^+^ system, such as H_2_ + C_2_H_5_^+^, CH_3_^+^ + CH_4_ and H + C_2_H_6_^+^, were considered, the reaction profile for H_3_^+^ + C_2_H_4_ → H_2_ + C_2_H_5_^+^ was not examined. Therefore, we initially constructed a full-dimensional potential energy surface for the H_3_^+^ + C_2_H_4_ → H_2_ + C_2_H_5_^+^ reaction using an ab initio quantum chemical calculation dataset obtained from on-the-fly preliminary dynamics simulation results. Monomial symmetrized approach (MSA2) software developed by Bowman and coworkers [21,22] was employed to fit the potential energies and their gradients on a permutational invariant polynomial basis. Details regarding the development of the potential energy surface (PES) are provided in the Methodology section. Subsequently, we present the computational outcomes of the rate coefficients for the H_3_^+^ + C_2_H_4_ → H_2_ + C_2_H_5_^+^ reaction on the developed PES employing classical molecular dynamic and ring polymer molecular dynamic (RPMD) methods [23,24,25,26], which represent the quantum mechanical behavior of nuclei by considering them as cyclic beads. As mentioned above, there is a substantial barrier in the C_2_H_5_^+^ → H_2_ + C_2_H_3_^+^ reaction [15], and the C_2_H_5_^+^ fragment has a relatively long lifetime [14]. Therefore, we evaluate the three-body dissociation process using the internal energy of C_2_H_5_^+^*.

## 2. Results and Discussion

### 2.1. Potential Energy Profile

Figure 1a,b display the potential energy profiles during the H_3_^+^ + C_2_H_4_ → H_2_ + C_2_H_5_^+^ and C_2_H_5_^+^ → H_2_ + C_2_H_3_^+^ reactions, respectively. In the H_3_^+^ + C_2_H_4_ reaction, the reactants directly formed the H_2_ + C_2_H_5_^+^ products through the C_2_H_5_^+^···H_2_ van der Waals well. Conversely, in the generation of H_2_ + C_2_H_3_^+^ products, there was a potential energy barrier between C_2_H_5_^+^ and the resulting H_2_ + C_2_H_3_^+^ fragments. Table 1 presents a comparison of the potential energies at the stationary points obtained from the analytical potential energy function and the MP2/cc-pVDZ data for the H_3_^+^ + C_2_H_4_ → H_2_ + C_2_H_5_^+^ reaction. As mentioned above, this sampling scheme does not cover the PES region for the C_2_H_5_^+^ → H_2_ + C_2_H_3_^+^ dissociation process, which will be discussed separately in terms of the vibrational energy distributions of C_2_H_5_^+^ generated from the H_3_^+^ + C_2_H_4_ → H_2_ + C_2_H_5_^+^ reaction. However, the C_2_H_5_^+^···H_2_ van del Waals well of the PES is relatively deeper than that in the MP2 results; our constructed PES yielded results that reasonably matched those from the MP2 calculations. The molecular structure of the C_2_H_5_^+^···H_2_ intermediate complex and key geometries for PES and MP2 are compared in Appendix A. Although the H_2_ bond direction relative to the CC bond differed from the MP2 results, the other geometric parameters were quite similar. In a barrierless ion–molecule reaction involving large exothermicity, the molecule has relatively higher rovibrational energies or a large kinetic energy release (KER). Therefore, the quality of the PES fit in this area is considered to have little effect on these results. Additionally, the MP2 results were compared with high-quality CCSD(T) results, showing consistent findings across all the datasets. While our MP2 results for the C_2_H_5_^+^ → H_2_ + C_2_H_3_^+^ reaction showed a slightly higher barrier compared to that of the previous MP2 calculations, they exhibited good correlation with the CCSD(T) results, as shown in Table 1. In a similar reaction, C_2_H_7_^+^ → H_2_ + C_2_H_5_^+^, the barrier height of the CCSD(T) was slightly higher than that of the DFT calculation [18]. It should be noted that previous studies indicated that the transition state barrier tends to be higher with MP2 than with CCSD(T) [27,28,29].

### 2.2. Rate Coefficients

The collision simulations were conducted using both the classical MD and RPMD methods over a temperature range from 100 to 300 K. The vibrational and rotational energies for the H_3_^+^ + C_2_H_4_ reactants, along with their relative translational energy, were thermalized at each temperature in the simulations. In this scenario, the thermal rate coefficient $k\left(T\right)$ was obtained using the following equation [30]:(1)$$\begin{array}{c} k\left(T\right)=\sqrt{\frac{8k_{B}T}{\pi \mu }}\pi b_{max}^{2}\frac{N_{r}}{N_{t}}\;, \end{array}$$
where $k_{B}$, μ, and $b_{max}$ represent the Boltzmann constant, the reduced mass, and the maximum impact parameter, respectively. The standard error $∆k\left(T\right)$ is given as follows:(2)$$\begin{array}{c} ∆k\left(T\right)=k\left(T\right){\left(\frac{N_{t}-N_{r}}{N_{t}N_{r}}\right)}^{\frac{1}{2}}\;, \end{array}$$
where *N_r_* and *N_t_* denote the number of reacted and total trajectories. Further details about the collision procedure are provided in the Methodology section below. Table 2 provides the essential details required for computing the rate coefficients. The parameter *N_r_* presented in Table 2 denotes the count of the trajectories resulting in H_2_ + C_2_H_5_^+^ products. Notably, in our simulations, no trajectories led to 2H_2_ + C_2_H_3_^+^ products, which is consistent with the anticipated high potential energy barrier in the C_2_H_5_^+^ → H_2_ + C_2_H_3_^+^ reaction (refer to Figure 1b). Consequently, it is anticipated that the H_2_ + C_2_H_3_^+^ products will form via an indirect process involving the vibrational excited C_2_H_5_^+^ fragment.

Figure 2 illustrates the thermal rate coefficients across temperatures ranging from 100 to 300 K. Our results from both the classical MD and RPMD simulations can be interpreted as the rate coefficients for the overall reaction, encompassing both the H_3_^+^ + C_2_H_4_ → H_2_ + C_2_H_5_^+^ and H_3_^+^ + C_2_H_4_ → H_2_ + H_2_ + C_2_H_3_^+^ reactions, as the H_2_ + C_2_H_3_^+^ products that were indirectly generated. At 300 K, both sets of our results are somewhat better than the experimental data (green scatter in Figure 2) provided in the KIDA database, but we reasonably reproduced the experimental results. Both the classical MD and RPMD rate coefficients decrease as the temperature decreases. This temperature dependency is similar to the coefficients observed in other ion–neutral reactions, such as H^−^ + C_2_H_2_ → H_2_ + C_2_H^−^ and H_3_^+^ + CO → H_2_ + HCO^+^/HOC^+^ reactions, as derived from the classical and RPMD simulations [30,31]. Notably, this temperature dependency cannot be captured using the temperature-independent Langevin rate equation, which is solely based on the charge-induced–dipole interaction. The rate coefficients obtained using RPMDs were slightly elevated compared to those obtained using classical MDs, suggesting that the quantum fluctuations in nuclei contribute to their increase. In RPMDs, ion–neutral attraction extends farther than the attraction between classical particles in classical MDs, owing to fluctuations in the nuclear probability densities. This observation aligns closely with findings from the H^−^ + C_2_H_2_ → H_2_ + C_2_H^−^ reaction [30]. In the following section, we will discuss the energy distribution of the H_2_ + C_2_H_5_^+^ product fragments resulting from proton affinity, aiming to comprehend the indirect mechanism of the C_2_H_5_^+^ → H_2_ + C_2_H_3_^+^ reaction.

### 2.3. Internal Energies of Fragments

To estimate the rovibrational energy distributions of the C_2_H_5_^+^ fragment concerning the indirect reaction C_2_H_5_^+^ → H_2_ + C_2_H_3_^+^, we examined the system internal energies (*ε*) of the C_2_H_5_^+^ fragment derived using the following equation:(3)$$\begin{array}{c} \varepsilon \;\left(C_{2}H_{5}^{+}\right)=E_{k}\;\left(C_{2}H_{5}^{+}\right)+\frac{1}{N_{bead}}\sum_{j=1}^{N_{bead}}\left\{V_{j}\left(\;{H_{2}+C}_{2}H_{5}^{+}\right)-\;V_{H_{2}}(r_{j})\right\}, \end{array}$$
where $E_{k}\;\left(C_{2}H_{5}^{+}\right)$ represents the vibrational and rotational energies for centroid velocities of the C_2_H_5_^+^ fragment, and $V_{j}\left(\;{H_{2}+C}_{2}H_{5}^{+}\right)$ denotes the potential energy obtained from the MSA2 PES concerning the nuclear configuration of *j*-th bead. Additionally, $V_{H_{2}}$ is a sixth-order polynomial function dependent on the internuclear distance (*r*) between the H_2_ fragments, where the center-of-mass distance (*R*) between the H_2_ and C_2_H_5_^+^ fragments is fixed at 20 Å. Figure 3a,b illustrate the relaxed potential energy curves as a function of *R* and *r*, respectively. Notably, no interaction was observed between the H_2_ and C_2_H_5_^+^ fragments when *R* exceeded 13 Å, as depicted in Figure 3a. Additionally, Figure 3b confirms that the sixth-order polynomial function effectively reproduced our PES for the C_2_H_7_^+^ system. For the final step, the trajectory data for *R* exceeding 13 Å are represented in the right-hand side of Equation (3). 

Figure 4 displays the *ε* of the C_2_H_5_^+^ fragment at temperatures of *T* = 100, 200, and 300 K. Notably, the peaks of *ε* using the classical MD and RPMD techniques at all the temperatures were below the asymptotic region (60.5 kcal/mol at the MP2 level) for the H_2_ + C_2_H_3_^+^ products, suggesting that the C_2_H_5_^+^ → H_2_ + C_2_H_3_^+^ dissociation channel was not dominant, although the distribution at *T* = 300 K indicates that the C_2_H_5_^+^ fragment with larger internal energies can lead to the H_2_ + C_2_H_3_^+^ dissociation channel. Both ε values decrease using the classical MD and RPMD techniques as the temperature declines, suggesting that the temperature-dependent behaviors of the internal energies of the H_3_^+^ and C_2_H_4_ reactants, along with their collision energy, influence the proton abstraction process of C_2_H_4_ from H_3_^+^. Consequently, we infer that the branching fraction for [C_2_H_5_^+^]:[C_2_H_3_^+^] is strongly temperature-dependent, potentially leading to the overestimation of the branching fraction of the C_2_H_3_^+^ product in previous experiments [13,14,17]. It is worth noting that the MSA2 PES was developed using the MP2 results for the dynamics simulations, and the C_2_H_5_^+^···H_2_ van del Waals well of the PES was relatively deeper than that in the MP2 results, as mentioned above. Further elaboration, along with quantitative results, can be expected by conducting theoretical dynamic simulations using a high-quality PES constructed with sophisticated techniques, such as the Δ-machine learning algorithm [32,33,34]. Moreover, as the temperature decreases, the RPMD distribution becomes narrower, indicating that at low temperatures, the quantum nuclei exhibit characteristic quantum behavior in the internal energy of C_2_H_5_^+^. It should be noted that the energy distributions were derived from a snapshot at the final step rather than from the Fourier transformation of the auto-correlation function.

Figure 5 illustrates the distributions of vibrational (*ε*_vib_) and rovibrational states (*ε*_rovib_) for the H_2_ fragment in the H_3_^+^ + C_2_H_4_ → H_2_ + C_2_H_5_^+^ reaction. The distributions were obtained using the following equations:(4)$$\begin{array}{c} \varepsilon_{vib}\;\left(H_{2}\right)=E_{k}^{vib}\;\left(H_{2}\right)+\frac{1}{N_{bead}}\sum_{j=1}^{N_{bead}}V_{H_{2}}(r_{j})\;, \end{array}$$
and
(5)$$\begin{array}{c} \varepsilon_{rovib}\;\left(H_{2}\right)=E_{k}^{vib}\;\left(H_{2}\right)+E_{k}^{rot}\;\left(H_{2}\right)+\frac{1}{N_{bead}}\sum_{j=1}^{N_{bead}}V_{H_{2}}(r_{j})\;, \end{array}$$
where $E_{k}^{vib}\;\left(H_{2}\right)$ and $E_{k}^{rot}\;\left(H_{2}\right)$ represent the vibrational and rotational energies for the centroid velocities of the H_2_ molecule. Unlike the distribution of the system’s internal energy for the C_2_H_5_^+^ fragment, the internal energy of H_2_, having fulfilled its role as a proton donor, is minimally affected by the temperature. As the temperature rises, the distributions for both *ε*_vib_ and *ε*_rovib_ expand due to the incorporation of thermal energies. The internal energies of classical MDs are notably low, primarily because the classical scheme does not account for zero-point energy. The peaks of the vibrational state distributions for RPMDs on the potential energy curve, accounting for anharmonicity, are approximately 4 kcal/mol. Notice that the zero-point energy of H_2_, which was calculated using the harmonic analysis of the MP2/cc-pVDZ level, is 6.4 kcal/mol.

The kinetic energy release (KER), which was not distributed along the internal modes, is depicted in Figure 6. These distributions were obtained using the following equations:(6)$$\begin{array}{c} E\;\left(KER\right)=\frac{1}{2}\mu_{rel}{\vec{v}}_{rel}^{2}, \end{array}$$
where $\mu_{rel}$ and ${\vec{v}}_{rel}$ denote the reduced mass and relative velocity vector between the H_2_ and C_2_H_5_^+^ fragments. The distributions broadened, reflecting the distribution of the internal energies of the H_2_ and C_2_H_5_^+^ fragments. However, no significant temperature dependence was observed in the relative energy for both the classical MD and RPMD results. It should be noted that the KER might have been underestimated because the exit channel for the H_2_ + C_2_H_5_^+^ products of our PES was relatively higher than the MP2 results.

## 3. Methodology

### 3.1. Development of a Global Potential Energy Surface

All the quantum chemistry computations to construct the full-dimensional C_2_H_7_^+^ PES were conducted at the MP2/cc-pVDZ level using Gaussian09 [35]. This level of calculation was chosen due to its ability to provide a substantial amount of energy and gradient data across various structures within a reasonable computational timeframe. This study uses a barrierless proton transfer reaction in the electronic singlet ground state. This proton transfer reaction, not involving electron transfer, has a relatively small electron correlation energy value. Table 1 shows that the MP2 energy is quite similar to the CCSD(T) results. Our previous dynamics study for the ion–molecule reaction, such as H^−^ + C_2_H_2_ → H_2_ + C_2_H^−^, qualitatively reproduced the rate coefficient [30]. It should be noted that for the NH_3_^+^ + H_2_ → NH_3_^+^ + H reaction, which involves a substantial barrier to hydride transfer, the CCSD(T) level was employed [36]. 

In the barrierless ion–molecule reactions, strong attractive forces like charge–dipole or charge-induced–dipole interactions play a crucial role. Therefore, quasi-classical trajectory calculations were computed for both the H_3_^+^ + C_2_H_4_ reactants and the H_2_ + C_2_H_5_^+^ products to sample the structural data points. A total of 220 trajectories from the reactant side and 60 trajectories from the product side with collision energies of 5, 10, and 20 kcal/mol were run, yielding 465,000 data points. All the trajectories from the reactants reached the H_2_ + C_2_H_5_^+^ products, without directly producing the 2H_2_ + C_2_H_3_^+^ product. Consequently, data sampling of the H_2_ + H_2_ + C_2_H_3_^+^ fragments was not performed. These data points were fitted to an analytical function comprising permutationally invariant polynomial basis sets using the MSA2 code developed by Bowman and coworkers [21,22]. The preliminary fit helped identify the unphysical leaky holes, which frequently occur in PES regions with insufficient data points [37]. An additional 2790 data points were added to fill these holes. The structures with higher energies, specifically those up to 0.2 hartrees from the C_2_H_5_^+^··· H_2_ complex, were subsequently excluded, reducing the number of data points to approximately 330,000. A fourth-order polynomial was employed, resulting in a final root-mean-square error of 959 cm^−1^ for the fit. Appendix A displays the plotted fitted potential energies relative to the MP2/cc-pVDZ energies. The fitted potential energy surface (PES) reasonably replicated the reaction energies for both the reactants and products, including the C_2_H_5_^+^··· H_2_ intermediate complex, along with the vibrational frequencies for this intermediate complex (see Figure 1 and Appendix A). For comparison with the MP2/cc-pVDZ results, the density fitting explicitly correlated coupled-cluster singles and doubles plus perturbative triples (DF-CCSD(T)-F12) method [38,39] with cc-pVDZ and cc-VTZ basis sets implemented using Molpro 2023.2 was utilized [40]. Global Reaction Route Mapping (GRRM) calculations [41,42,43] were performed to obtain the equilibrium and transition state structures.

### 3.2. Procedure for Molecular Dynamics

All the nuclear dynamics simulations conducted in this study were based on the fitted PES. Initially, path integral molecular dynamic (PIMD) simulations were executed to derive the initial coordinates and momenta for collisional simulations spanning temperatures from *T* = 100 K to 300 K. The ring polymer Hamiltonian $H\left(p,r\right)$ [30] is defined as follows:(7)$$\begin{array}{c} H\left(p,r\right)=\sum_{i=1}^{3n}\left\{\frac{1}{N_{bead}}\sum_{j=1}^{N_{bead}}\frac{{\left(p_{i}^{j}\right)}^{2}}{2m_{i}}+\frac{m_{i}N_{bead}}{2\beta^{2}\hbar^{2}}\sum_{j=1}^{N_{bead}}{\left(r_{i}^{j}-r_{i}^{j-1}\right)}^{2}\right\}+\frac{1}{N_{bead}}\sum_{j=1}^{N_{bead}}V\left(r_{1}^{j},\ldots ,r_{3n}^{j}\right), \end{array}$$
where ℏ and β represent the reduced Planck constant and reciprocal temperature, respectively, where $\beta \equiv 1/k_{B}T$. $m_{i}$ stands for the atomic mass of the *i*-th nucleus, and V signifies the potential energy of the system. $N_{bead}$ represents the number of beads, while $r_{i}^{j}$ and $p_{i}^{j}$ denote the Cartesian coordinates and their conjugated momentum vectors of the *j*-th bead for the *i*-th atom, respectively. Utilizing the ring polymer Hamiltonian [30] defined by the PIMD method, phase information was obtained in accordance with the quantum Boltzmann distribution, effectively capturing the nuclear quantum effects, including discretized vibrational energies with an appropriate number of beads (*N*_bead_). The convergence of *N*_bead_ is demonstrated in Appendix A, illustrating its association with the system’s internal energy. In this study, RPMD simulations were conducted with varying numbers of beads, *N*_bead_ = 96, 64, 48, 48, and 24 for temperatures *T* = 100, 150, 200, 250, and 300 K, respectively. To maintain the configurations where the C_2_H_4_ and H_3_^+^ reactants do not interact, a harmonic bias potential was employed alongside a Nosé–Hoover thermostat for canonical ensembles. The integration of the equations of the motion of the ring polymer Hamiltonian was carried out using the velocity–Verlet method with a time step of Δ*t* = 0.10 fs, totaling 10^4^–10^6^ simulation steps. Subsequently, RPMD simulations were performed, extending the PIMD method to enable real-time dynamics simulations, which are particularly adept at capturing nuclear quantum effects, such as zero-point energy and tunneling [30,36,44,45,46,47,48,49]. The impact parameter (*b*) for collisional simulations was set below *b* = *b_max_* *ζ*^1/2^, where *b_max_* represents the maximum impact parameter, and *ζ* is a random number within the range of [0, 1]. Following this, the reactant coordinate was adjusted based on the specified impact parameter. Further specifics regarding the collisional simulation can be found in our previous publication [30]. Eventually, the RPMD trajectories were propagated by solving the equations of the motion of the ring polymer Hamiltonian without a thermostat, employing a time step of Δ*t* = 0.10 fs. The total simulation time spanned 1.5–22 ps. Additionally, classical MD simulations were conducted for comparison with the RPMD results, following a similar procedure except for the number of beads, which remained constant at one for all the temperatures in the classical MD simulations. All the calculations for PIMDs, RPMDs, and classical MDs were performed using the open-source code *PIMD.ver.2.6.0* [50].

## 4. Conclusions

The thermal rate coefficients for the overall reaction, encompassing both the direct reaction H_3_^+^ + C_2_H_4_ → H_2_ + C_2_H_5_^+^ and the indirect reaction C_2_H_5_^+^ → 2H_2_ + C_2_H_3_^+^, were determined across a temperature range from 100 to 300 K. The coefficients exhibited an increase as the temperature increased. The coefficient at 300 K was slightly higher than the value in the KIDA database, but we still reasonably reproduced it. Comparatively, the rate coefficients obtained from the RPMD simulations showed a slight increase compared to those from the classical MD simulations, which were attributed to fluctuations in the nuclear probability densities [30]. Moreover, in order to explore the indirect mechanism of the H_3_^+^ + C_2_H_4_ → H_2_ + C_2_H_5_^+^ → 2H_2_ + C_2_H_3_^+^ reaction, we analyzed the distributions of internal energies for the H_2_ + C_2_H_5_^+^ fragments and the relative translational energy (*E* (KER)). Despite the influence of the proton affinity of H_3_^+^ on generating the C_2_H_5_^+^ fragment, its internal energies were insufficient to reach the dissociation limit of the H_2_ + C_2_H_3_^+^ products. While the *ε* of C_2_H_5_^+^ fragment decreased with decreasing temperature, the internal energy of H_2_ and the *E* (KER) exhibited a weak correlation with the temperature. Based on these findings and the estimation of the internal energies of the C_2_H_5_^+^ fragments, we suggest that the H_3_^+^ + C_2_H_4_ → H_2_ + C_2_H_5_^+^ reaction prevailed, while the C_2_H_5_^+^ → 2H_2_ + C_2_H_3_^+^ reaction was of lesser significance. Here, it should be mentioned that the comparison of theoretical rate coefficients at 300 K with the experimental results and the temperature dependency of the *ε* of C_2_H_5_^+^ fragment was investigated qualitatively using the developed PES. In the future, we recommend an experimental investigation into the temperature dependency of the branching fraction for [C_2_H_5_^+^]:[C_2_H_3_^+^], complemented by theoretical results based on a high-quality PES constructed using sophisticated techniques such as the Δ-machine learning algorithm [32,33,34].

## Figures and Tables

**Figure 1 molecules-29-02789-f001:**
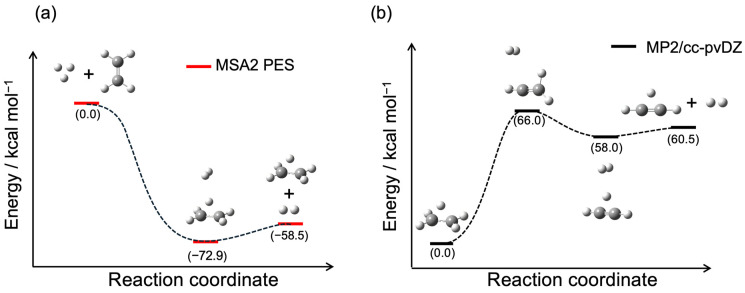
Diagram of reaction coordinates, illustrating the geometries at stationary points for (**a**) the H_3_^+^ + C_2_H_4_ → H_2_ + C_2_H_5_^+^ reaction obtained using MSA2 PES and (**b**) the C_2_H_5_^+^ → H_2_ + C_2_H_3_^+^ reaction calculated at the MP2/cc-pVDZ level. Zero energy is defined as the H_3_⁺ + C_2_H_4_ reactant energy level in (**a**), while it is defined as the C_2_H_5_⁺ energy level in (**b**).

**Figure 2 molecules-29-02789-f002:**
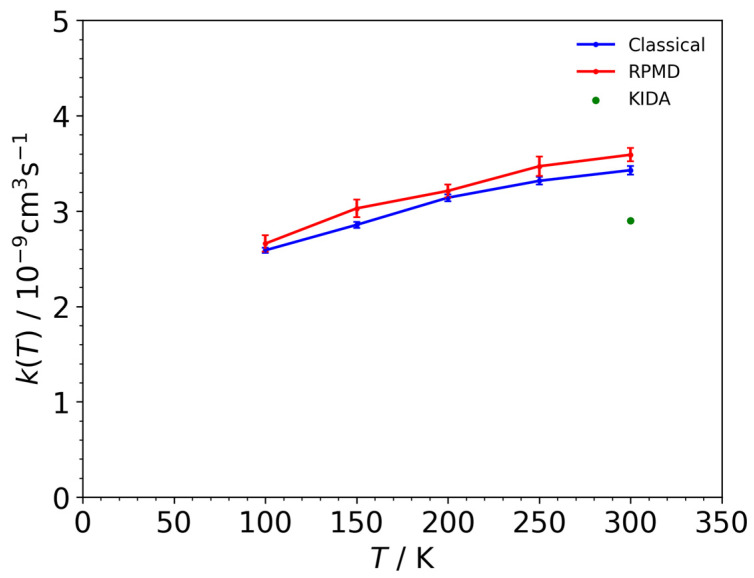
The thermal rate coefficients for the overall reactions, encompassing both H_3_^+^ + C_2_H_4_ → H_2_ + C_2_H_5_^+^ and H_3_^+^ + C_2_H_4_ → 2H_2_ + C_2_H_3_^+^, derived from the classical MD results (blue line) and RPMD outcomes (red line). The green scatter plot denotes the rate coefficients extracted from the KIDA database for the overall reactions.

**Figure 3 molecules-29-02789-f003:**
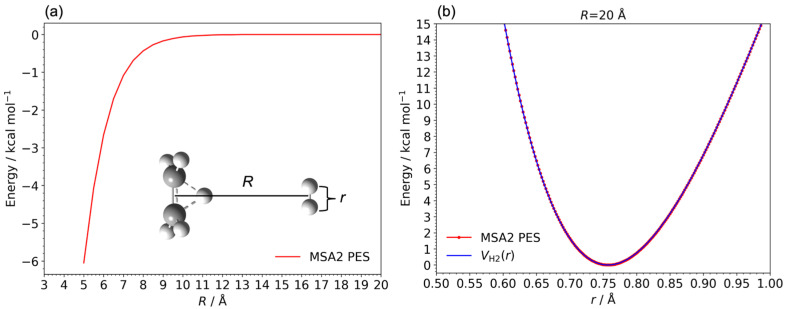
The potential energy curves (red lines) of the C_2_H_7_^+^ system as a function of (**a**) the center-of-mass distance (*R*) between the H_2_ and C_2_H_5_^+^ fragments and (**b**) the internuclear distance (*r*) between the H_2_ fragments, respectively. The molecular structures of *R* and *r* are depicted as insets in Figure 3a. The blue lines in Figure 3b represent the 6th order polynomial function dependent on *r*. The zero energies are defined at the potential energy of the asymptote region between H_2_ and C_2_H_5_^+^.

**Figure 4 molecules-29-02789-f004:**
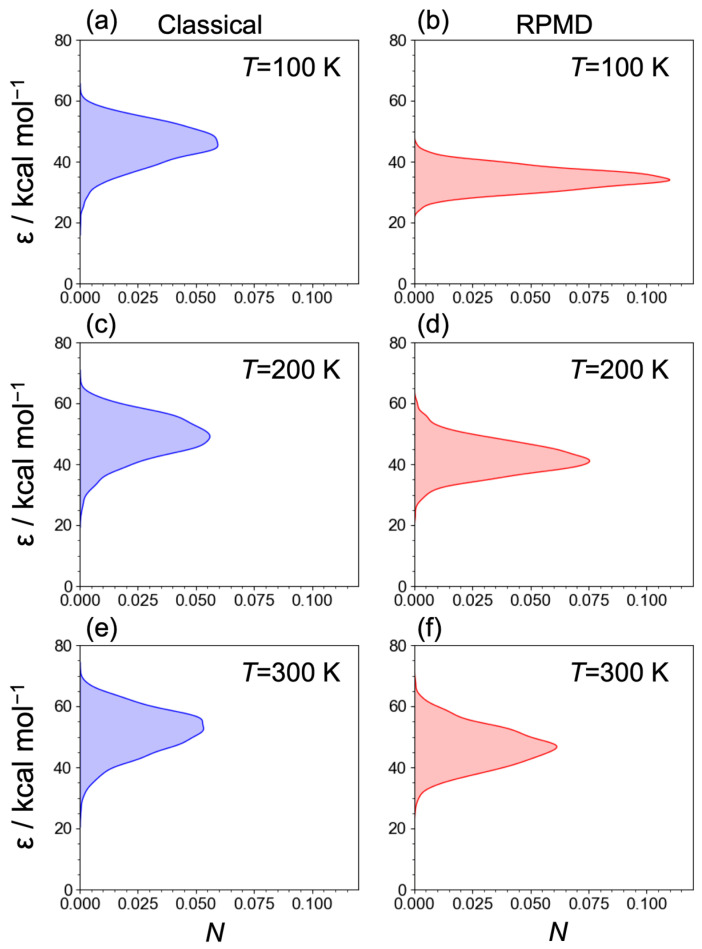
The internal energies (*ε*) of the C_2_H_5_^+^ fragment for (**a**) classical MDs at *T* = 100 K, (**b**) RPMDs at *T* = 100 K, (**c**) classical MDs at *T* = 200 K, (**d**) RPMDs at *T* = 200 K, (**e**) classical MDs at *T* = 300 K, and (**f**) RPMDs at *T* = 300 K.

**Figure 5 molecules-29-02789-f005:**
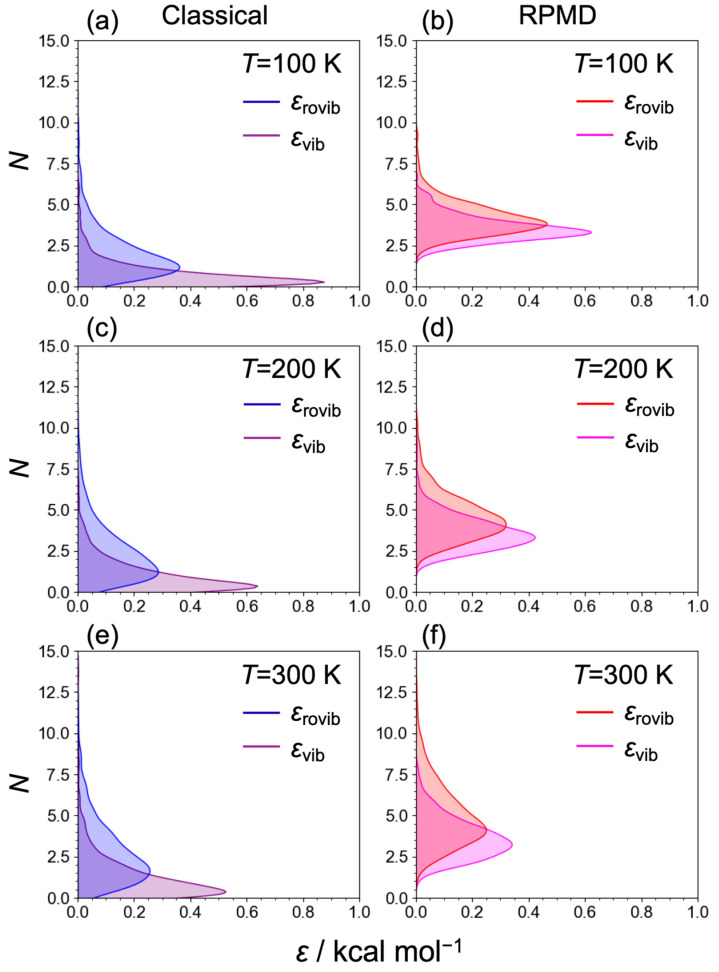
The vibrational (*ε*_vib_) and the rovibrational states (*ε*_rovib_) of the H_2_ fragment for (**a**) classical MDs at *T* = 100 K, (**b**) RPMDs at *T* = 100 K, (**c**) classical MDs at *T* = 200 K, (**d**) RPMDs at *T* = 200 K, (**e**) classical MDs at *T* = 300 K, and (**f**) RPMDs at *T* = 300 K. The purple and blue plots depict *ε*_vib_ and *ε*_rovib_ for classical MDs, whereas the magenta and red illustrate *ε*_vib_ and *ε*_rovib_ for RPMDs.

**Figure 6 molecules-29-02789-f006:**
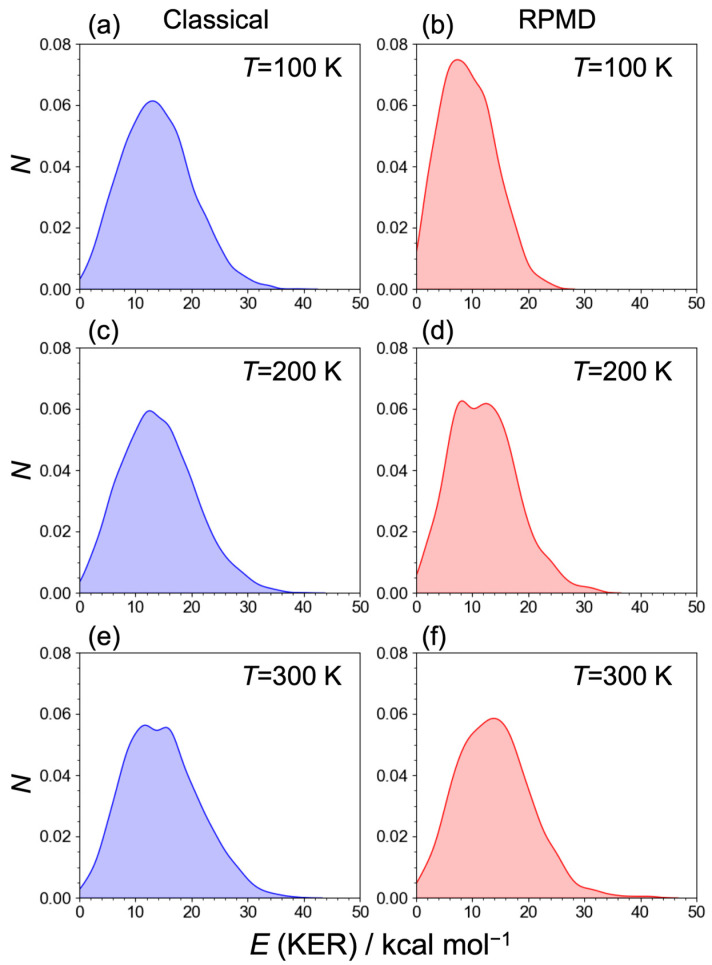
Kinetic energy release *E* (KER) corresponding to the relative translational energy between H_2_ and C_2_H_5_^+^ fragments for (**a**) classical MDs at *T* = 100 K, (**b**) RPMDs at *T* = 100 K, (**c**) classical MDs at *T* = 200 K, (**d**) RPMDs at *T* = 200 K, (**e**) classical MDs at *T* = 300 K, and (**f**) RPMDs at *T* = 300 K.

**Table 1 molecules-29-02789-t001:** Relative energies for reactions (**a**) and (**b**) for MSA2 PES, MP2/cc-pVDZ, DF-CCSD(T)-F12a/cc-pVDZ, and cc-pVTZ, as well as MP2/6-31G(d,p). The zero energies are defined by the potential energy of the reactant.

	PES	MP2	DF-CCSD(T)-F12a		MP2
		cc-pVDZ	cc-pVDZ	cc-pVTZ	6-31G(d,p) *^a^*
(**a**) H_3_^+^ + C_2_H_4_ → [C_2_H_5_^+^···H_2_] → H_2_ + C_2_H_5_^+^					
Reactant	0.0	0.0	0.0	0.0	—
Intermediate complex	−72.9	−65.2	−65.3	−63.8	—
Product	−58.5	−63.7	−63.6	−62.0	—
(**b**) C_2_H_5_^+^ → [C_2_H_3_^+^···H_2_]^‡^ → [C_2_H_3_^+^···H_2_] → H_2_ + C_2_H_3_^+^					
Reactant	—	−63.7 (0.0)	−63.6 (0.0)	−62.0 (0.0)	(0.0)
Transition state	—	2.3 (66.0)	−0.2 (63.4)	−0.6 (61.4)	(55.9)
Intermediate complex	—	−5.7 (58.0)	−4.2 (59.4)	−5.3 (56.7)	—
Product	—	−3.2 (60.5)	−1.5 (62.1)	−2.6 (59.4)	(50.0)

*^a^* Data from Ref. [15].

**Table 2 molecules-29-02789-t002:** Thermal rate coefficients with the standard errors (*k*(*T*) ± Δ*k*(*T*)) estimated from classical MD and RPMD results, along with the impact parameter (*b_max_*) and the number of reacted (*N_r_*) and total trajectories (*N_t_*), as well as the coefficients in the KIDA database.

	Classical				RPMD				KIDA
*T* (K)	(*k*(*T*) ± Δ*k*(*T*)) × 10^−9^	*b_max_* (Å)	*N_r_*	*N_t_*	(*k*(*T*) ± Δ*k*(*T*)) × 10^−9^	*b_max_* (Å)	*N_r_*	*N_t_*	*k*(*T*) × 10^−9^
100	2.59 ± 0.03	12.0	3245	4994	2.66 ± 0.09	13.5	453	859	-
150	2.86 ± 0.03	11.5	3183	4995	3.03 ± 0.09	13.0	514	972	-
200	3.14 ± 0.04	11.5	3028	4989	3.21 ± 0.07	12.5	1036	1972	-
250	3.31 ± 0.04	11.5	2856	4981	3.47 ± 0.10	12.0	531	964	-
300	3.43 ± 0.04	11.5	2703	4996	3.59 ± 0.07	11.5	1119	1975	2.9

## Data Availability

The data are contained within the article.

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
