# Peer review of "Theoretical Study of the Thermal Rate Coefficients of the H3+ + C2H4 Reaction: Dynamics Study on a Full-Dimensional Potential Energy Surface"

_molecules, 2024, doi:10.3390/molecules29122789_

Round 1

Reviewer 1 Report

Comments and Suggestions for Authors

The paper presents a potential energy surface based on a fitting to a permutationally invariant polynomial (PIP) model. The adjustment is made using neural-network techniques, therefore the surface can be referred to as PIP-NN PES. For the fit, the authors perform a dot grid based on 330,000 energy calculations at the MP2/cc-pVDZ level. With the developed surface, RPMD and classical MD simulations are performed to obtain the velocity constant. The paper is well-structured and the results are well presented.

However, it would be necessary to make the following considerations to the authors:

1. These types of surfaces have been widely developed by various research groups, from the works of Bowman (referenced by the author and using his code), to the works of Hua Guo, Dong Zhang or Gábor Czakó, among others. However, unlike the work of the previous groups, in the title, the authors indicate that it is a "machine-learning" potential energy surface rather than identifying it as a permutationally invariant polynomial neural-network (PIP-NN) PES. The term "machine-learning" in the title can be misleading, as it appears that the potential energy surface is a machine learning-based model that is capable of predicting the energy of the system. The title of the paper should not include the term "machine learning" and should simply be referred to as PIP-NN PES.

 2.      In the abstract, (page 1, line 15) for clarity, as in the title, when it is indicated that a "machine learning approach" is used, this sentence should indicate that a potential energy surface based on a fitting is developed, using a machine learning approach for fitting to a polynomial permutationally invariant function. In the same way, the use of the term machine-lerning on page 2, line 72, should be better explained.

 3.      Page 3-4. Potential energy profile. As can be seen in Table 1, the comparison between the values at the MP2 level and CCSD(T)-F12a/cc-pVDZ and cc-pVTZ are adequate, less than 2 kcal/mol, taking into account that the MP2 levels always overestimate the energy. However, the values obtained with the PES are very different, 7 kcal/mol for the intermediate complex and 5 kcal/mol for the products in the reaction (a). Could it be an error in the training dataset fiiting? As can be seen later, the mean square error is 959 cm−1 (2.74 kcal/mol).

 4.      Page 6-7. 2.3. Internal energy of fragments. In line 205, the authors acknowledge the limitations of the surface because the training dataset has been obtained at the MP2 level and that higher-level calculations would be needed to obtain quantitative results. At the same point, in line 209 the authors refer to the ∆-machine learning methodology. This methodology can be considered a machine learning surface, since a set of high-level energy data obtained through a single point is used to train a model that makes predictions for the rest of the geometries of the dataset that is later used to adjust the surface.

 5.      Page 10. 3. Methodology. In line 267 the authors indicate that a total of 330,000 points are obtained in total, however, in line 272 they indicate that, after excluding structures that move more than 0.2 hartree, it is still approximately 330,000 points. What percentage of points have been removed? It’s a mistake?

 6.      In section 3.2. The convergence study of the number of beads carried out by the authors  (which appears in figure S2 of the supplementary material) is to be appreciated, however, I do not find in the paper the number of beads that has finally been used to perform the RPMD simulations.

 7.      Page 11. 4. Conclusions. Despite the fact that at the end of the conclusions the authors indicate the need to improve the surface with higher level calculations and the use of ∆-machine learning, it is not indicated that the results obtained should be taken into account from a qualitative and not a quantitative point of view due to the uncertainty introduced by the MP2 level in obtaining the energies of the points fitted.

Reviewer 2 Report

Comments and Suggestions for Authors

In this study, the authors investigate the  H3+ + C2H4 H2 + C2H5+ / 2H2 + C2H3+ reaction using the machine-learning method based on MP2/cc-pVDZ. However, the results significantly contradict with the experimentally observed 0.30:0.70 branching fraction ration for [C2H5+]:[C2H3+]. The theoretical study completely miss the indirect reaction. These results clearly indicated that the selected method is not suitable for this reaction. Moreover, in the methodology section, some details are missed for the development of global potential energy surface (PES). It is not clear how many points have been computed by the MP2 method and how to construct the PES based from these MP2 results.  In my personal opinion, I think it should be that MP2 is not suitable for this system. It is also confused for me that the energy of the intermediate is lower than that of the product in Figure 1. Therefore, I can't recommend the publication of this manuscript.

Reviewer 3 Report

Comments and Suggestions for Authors

In the manuscript, the authors constructed the full-dimensional potential energy surface for the C2H7+ system, and dynamics processes of H3+ + C2H4 H2 + C2H5+ and C2H5+ H2 + C2H3+ are studied by using classical molecular dynamics and ring-polymer molecular dynamics methods. The calculated reaction rate coefficient value at 300 K agrees well with the values in the kinetic database, and validating the accuracy of the new constructed PES. In addition, the energy distribution of the C2H5+ fragments are studied. As a whole, this work is meaningful and scientific, I suggest this work can be published after a minor revision.

1. In this paper, the collision between H3+ and C2H4 is studied, mainly for the two channels H2 + C2H5+ and 2H2 + C2H3+. However, the branching ratio of the two channels are not in good agreement with the values in the KIDA database. So, can the authors discuss the reason for this result from the perspective of PES features? In addition, are there any other reaction channels for the H3+ + C2H4?

2. In the Methodology part, when choosing molecular configurations using quasi-classical calculations, only the H3+ + C2H4 and H2 + C2H5+ channels are considered, why not also the 2H2 + C2H3+ channel?

3. In fitting the energy point data, the authors used the permutationally invariant polynomial method with MSA2 code, but saw no introduction to machine learning method related to this work.

Reviewer 4 Report

Comments and Suggestions for Authors

This work explores how H3+ can influence ethylene chemistry in the gas phase.  This is a fairly straightforward reaction with fairly simple molecular systems.  The Machine Learning component feels like a gimmick and doesn’t really contribute to the science; in some ways (like Figure 1), the Machine Learning presents a wrong conclusion taking away from the outcomes.  My overall estimate is that this work is a simple system with a simple analysis with straightforward tools that is not really pushing science strongly forward.  Yes, a criticism of the KIDA database is raised (which is potentially useful), but this current manuscript is weak.  Additionally, the organization is strange in places (discussed below), and the narrative is difficult to follow as a result.  If the methods employed were more accurate, higher-level, or showed an advancement of Machine Learning techniques on reaction chemistry, I would be more enthusiastic in my support.  However, I remain tepid at best.

Only the first paragraph of the introduction is actually introduction.  The rest is a discussion of the current results which is not fitting.  Some introductory material is present in lines 73-79, but this is mostly methodology.  Hence, the current text is grossly lacking in introduction and motivation.

MP2 is not sufficient for any reaction profile or for training data, but the authors benchmark this against CCSD(T)-F12a which, I guess, is acceptable.  However, the relatively small number of electrons should make this a straightforward case for utilization of CCSD(T)-F12 computations potentially including all kinds of additionally-helpful corrections like core electrons, higher-order correlation, etc.

Similarly to the previous point, the first 3 sentences of the Conclusions don’t really offer anything except a restatement of the methods.  This really isn’t useful.  In this present style, it’s really difficult for me to adjudicate the value of the present research results.  What was actually found?  For instance, the lack of support for a direct creation of H2 + C2H5+ in this reaction is a noteworthy result, but it’s buried in the middle of the Conclusions making it difficult for a casual reader to see the value produced herein.

Comments on the Quality of English Language

The language can be improved some with copy editing or similar.

Round 2

Reviewer 2 Report

Comments and Suggestions for Authors

The author's replies could not dispel my concerns about this article and I still can not recommend the publication of this manuscript. 

Reviewer 4 Report

Comments and Suggestions for Authors

The updated work by Murakami is an improvement over the previous version.  The authors have tried to address my concerns and have done so sufficiently.  I still feel like the computational approach could be better, but the authors make valid arguments for their method.  They have also improved the discussion in the Introduction and moved the most important result to the front of the Conclusions.  Hence, I support publication of this work after a little English editing by the MDPI staff.

Comments on the Quality of English Language

I support publication of this work after a little English editing by the MDPI staff.
